# Comparable Post-Vaccination Decay of Neutralizing Antibody Response to Wild-Type and Delta SARS-CoV-2 Variant in Healthcare Workers Recovered from Mild or Asymptomatic Infection

**DOI:** 10.3390/vaccines10040580

**Published:** 2022-04-09

**Authors:** Ilaria Vicenti, Monica Basso, Filippo Dragoni, Francesca Gatti, Renzo Scaggiante, Lia Fiaschi, Saverio G. Parisi, Maurizio Zazzi

**Affiliations:** 1Department of Medical Biotechnologies, University of Siena, Viale Bracci 16, 53100 Siena, Italy; vicenti@unisi.it (I.V.); dragoni16@student.unisi.it (F.D.); lia300790@gmail.com (L.F.); maurizio.zazzi@unisi.it (M.Z.); 2Department of Molecular Medicine, University of Padova, Via Gabelli, 63, 35100 Padova, Italy; monica.basso@unipd.it (M.B.); francesca.gatti@asst-garda.it (F.G.); 3Belluno Hospital, Viale Europa, 22, 32100 Belluno, Italy; renzo.scaggiante@aulss1.veneto.it

**Keywords:** COVID-19, vaccination, Delta, BNT162b2 vaccine, neutralizing antibody

## Abstract

We described the long-term decay of neutralizing antibody (NtAb) to the wild-type and Delta SARS-CoV-2 variant after three antigen stimulations (mild or asymptomatic natural infection followed by two doses of the BNT162b2 mRNA vaccine after a median of 296 days) in immunocompetent healthcare workers (HCWs). Live virus microneutralization against the B.1 and Delta SARS-CoV-2 variants was performed in VERO E6 cell cultures. The median NtAb titers for B.1 and Delta were comparable and highly correlated at both 20 and 200 days after the second vaccine dose in the 23 HCWs enrolled (median age, 46 years). A small group of naturally infected unvaccinated HCWs had comparable NtAb titers for the two strains after a median follow-up of 522 days from infection diagnosis. The NtAb response to the Delta VoC appears to follow the same long-term dynamics as the wild-type response regardless of the vaccinal boost; data collected after three antigen stimulations (natural infection followed by two doses of the BNT162b2 mRNA vaccine) may be helpful for tailoring the continuous monitoring of vaccine protection against SARS-CoV-2 variants over time.

## 1. Introduction

While massive vaccination against SARS-CoV-2 has been consistently pursued worldwide, the emergence and spread of new virus variants is continuously challenging the efficacy of both natural and artificial immunization. Until December 2021, two largely dominant variants of concern (VoCs) had spread widely in different waves of the epidemics. The Alpha variant quickly replaced the wild-type SARS-CoV-2 lineages and dominated from December 2020 to June 2021; then, the Delta variant quickly superseded the Alpha variant, remaining the largely prevailing lineage until December 2021. Notably, all the administered vaccines are based on the wild-type Wuhan-like B.1 isolate; thus, cross-protection through vaccination remains a key point to monitor.

Extensive work demonstrated that vaccination-elicited antibodies can neutralize the subsequently emerging VoCs, including alpha, beta and gamma, in vitro, although with reduced titers compared to the immunizing B.1 lineage [1,2,3]. Published data indicate a similar picture for cross-protection against the currently dominating Delta VoC, with 2.5- to 20-fold-decreased titers of neutralizing antibody, depending on the vaccine preparation and timing of assessment with respect to the vaccine schedule [4,5,6,7]. However, few data about the decay of neutralizing antibody (NtAb) to the Delta variant following infection and/or vaccination are available.

The aim of this study was to describe the long-term decay of NtAb to the wild-type strain and Delta variant in a cohort of healthcare workers (HCWs) vaccinated with two doses of the BNT162b2 mRNA vaccine after asymptomatic or mild natural infection.

## 2. Materials and Methods

The study cohort included immunocompetent HCWs who were infected from March to April 2020 in Northern Italy: the diagnosis was made by the detection of SARS-CoV-2 RNA in the upper respiratory airways by molecular testing, and the participants received supportive care only prescribed by an infectious disease specialist. Two HCWs had uncomplicated arterial hypertension. The subjects were vaccinated with the BNT162b2 mRNA vaccine, and they received the second dose 3 weeks after the first dose. No allergic responses to the vaccinations occurred. All the HCWs underwent regular surveillance: no reinfections occurred.

Their NtAb levels were tested 20 days (T20) and 200 days (T200) after the second vaccine dose. Live virus microneutralization against the B.1 (EPI_ISL_2472896) and Delta (EPI_ISL_2840619) SARS-CoV-2 variants was performed in VERO E6 cell cultures as previously described [8]. The serum neutralization titer (ID_50_) was defined as the reciprocal value of the serum dilution that showed a 50% protection of virus cytopathic effect. Antibodies with ID_50_ titers ≥10 were defined as SARS-CoV-2 seropositive and neutralizing; sera with ID_50_ < 10 were defined as negative and scored as 5 for statistical analysis.

The non-parametric Wilcoxon signed rank-sum test and Mann–Whitney test were used to analyze the changes in paired and unpaired data, respectively. Spearman analysis was used to measure the correlation between the NtAb titers for the two viral strains.

The study was approved by the comitato per la sperimentazione clinica di Treviso e Belluno (protocol code 812/2020), and written informed consent was obtained from all the patients enrolled. The study was conducted according to the guidelines of the Declaration of Helsinki.

## 3. Results

A total of HCWs (6 males, median (IQR) age 46 (40–52) years) were enrolled. Out of these, 13 were asymptomatic (M/F, 3/10), and 10 had mild disease (M/F, 5/5). Most of the HCWs with mild disease had fevers (8 out of 10). The median interval between the diagnosis of natural infection and the first vaccine dose was 296 (286–303) days, and the HCWs were all vaccinated in a 1-month period. At 20 days, the median (IQR) NtAb titers against the Delta virus were lower than the titers against the wild type (1432.0 (1188.9–2277.6) vs. 1837.6 (1449.2–2426.7)), but the difference was not statistically significant (*p* = 0.3070).

At 200 days, the median (IQR) wild-type and Delta NtAb titers remained comparable (442.0 (308.7–855.5) vs. 436.9 (247.8–673), *p* = 0.6057), and the positive correlation was confirmed (rho, 0.804; *p* < 0.0001).

Both the NtAbs to the wild-type and Delta forms decreased significantly from day 20 to day 200 (*p* < 0.0001). A detailed description is reported in Figure 1 and Figure 2.

Interestingly, the median NtAb titers for the wild type and Delta were also comparable (30.5 (18.5–46.5) vs. 28.3 (22.1–37.7), *p* = 1) in four naturally infected unvaccinated HCWs (Figure 3): this result suggests that a comparable response to these different variants is independent from vaccination.

## 4. Discussion

These findings confirm the waning of the NtAb response over time, despite a natural infection plus two-dose vaccination, as shown in a large study of vaccinated HCWs [9]. However, the NtAb response to the Delta VoC appears to follow the same dynamics as that to the wild-type immunogen included in all the current vaccines. The lack of long-term SARS-CoV-2 immunity may explain the surge of SARS-CoV-2 Delta infections in vaccinated people, but there are other hypotheses, as a role for Delta-specific escape from vaccine-induced immunity need to be taken into account: mutations in the main antigenic sites of the Delta spike have the ability to evade recognition by monoclonal antibodies [10]. The different genomic characteristics of VoCs have clinical relevance: many of the Omicron RBD mutations are found at positions which are important contact sites with ACE2, and Cameroni et al. [11] reported that the Omicron RBD binds to human ACE2 with increased affinity.

The Delta VoC was first documented in India in December 2020. This VoC presents a unique mutation, T478K, in the spike glycoprotein with a high binding affinity for ACE2; along with L452R, it facilitates immune escape [12,13,14]. The clinical relevance is related to the faster replication, the higher contagiousness before symptom onset and the occurrence of superspreading events, which make contact tracing more difficult [15,16]. Furthermore, the hospitalization rates for Delta are higher than those for pre-Delta variants [17]. The level of neutralizing antibodies reflects the humoral immune response against SARS-CoV-2 infection, and the BNT162b2 vaccine has a very large effect on it [18]; however, both in the case of natural infection elicited by the wild-type strain and in the case of vaccination, VoCs may evade immune responses. For Delta, which bears key immune-escape mutations such as L452R, P681R, and T478K, a substantial fold reduction in the neutralizing antibodies elicited by vaccination or infection compared to that for reference lineages was previously reported [19].

We included in the study previously infected subjects tested 20 and 200 days after the second BNT162b2 mRNA vaccine; a detectable level of NtAb at the longest interval, similar to our findings, is expected [20,21]. Establishing a threshold for in vivo protection is a complex issue because several factors must be considered, including T-cell immunity and containment measures as well as selection biases impacting exposure to the virus in different population studies. A recent study defined the median threshold for in vivo protection at 154 binding antibody units (BAUs) in a spike RBD-binding assay with the ancestral wild-type virus [22]; however, establishing correspondence with the different neutralization assays remains tricky.

The BNT162b2 vaccine estimated efficacy after a two-dose regimen was up to 99% for the prevention of death and 100% for the prevention of severe COVID-19 [23]; head-to-head comparisons showed a slightly lower protective effect for symptomatic disease or any infection caused by Delta compared to Alpha [24]. The spike protein is responsible for binding to host receptors, a key function driving evolution towards highly human-adapted phenotypes via point mutations and recombination events occurring at higher rates here than in any other part of the viral genome [25,26].

Many VoCs arose at the end of 2020, with mutations spanning the entire protein but mostly in S1 and RBD, which is the main target of neutralizing antibodies. However, non-RBD mutations are also involved in escape from vaccine-induced humoral immunity, and the continuous monitoring of the emergence of new variants is warranted, together with the detection of breakthrough infections [27,28]. Several previously published studies reported a lower vaccine-induced cross-neutralizing activity against the Delta variant compared to the wild-type strain—our apparently contrasting results may have been influenced by the low number of HCWs tested—thus, further studies are needed to confirm our data.

## 5. Conclusions

At present, the third vaccine doses are being administered to all people as a booster, and no updated vaccine preparations are available: long-term data on after three antigen stimulations (mild or asymptomatic natural infections followed, after a definite long interval, by two doses of the BNT162b2 mRNA vaccine) may be helpful for tailoring the continuous monitoring of vaccine protection against SARS-CoV-2 variants over time

## Figures and Tables

**Figure 1 vaccines-10-00580-f001:**
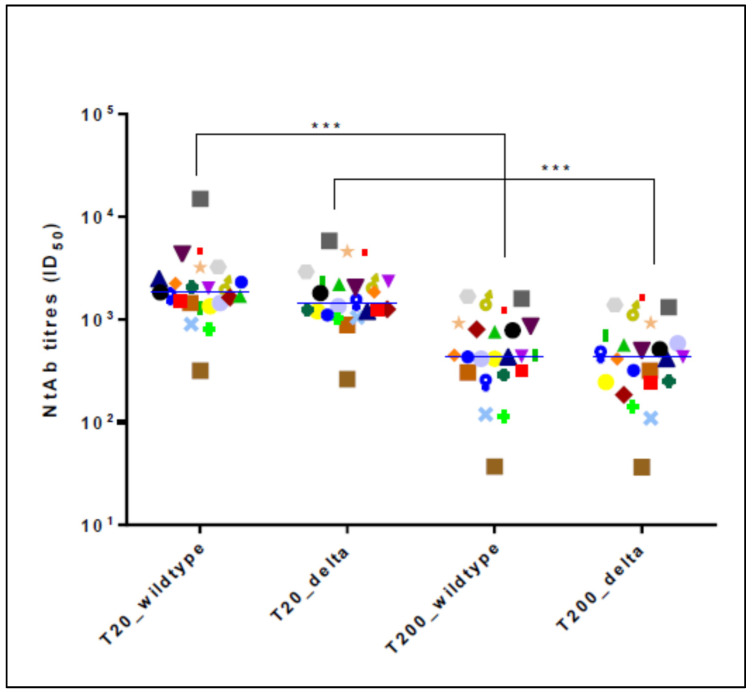
Neutralizing antibody titers for the wild-type lineage and the Delta variant in previously infected healthcare workers tested at 20 (T20) and 200 (T200) days following the second dose of the BNT162b2 vaccination. Data are reported as individual ID_50_ values and as median values at each study time point. The same-colored symbols indicate the individual ID_50_ values for the same subject at different time points: *** *p* < 0.001. ID_50_: reciprocal value of the serum dilution that showed a 50% protection of virus cytopathic effect.

**Figure 2 vaccines-10-00580-f002:**
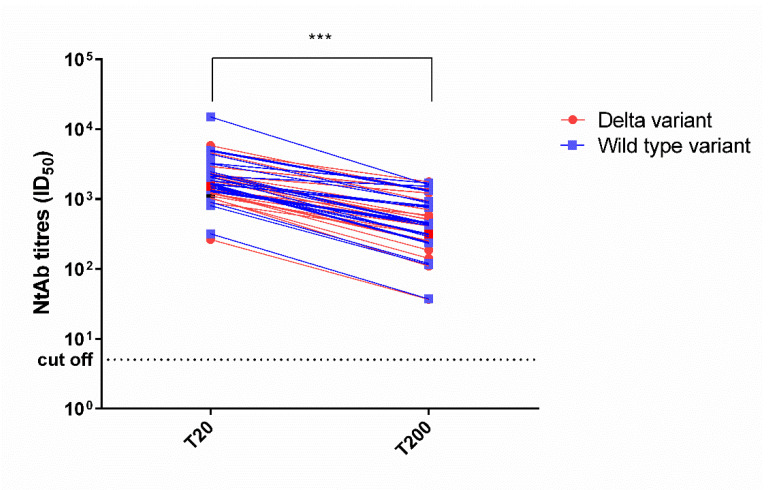
Longitudinal course and individual ID_50_ values of neutralizing antibody titers at 20 (T20) and 200 (T200) days following the second dose of the BNT162b2 vaccination in previously infected healthcare workers. The colored symbols indicate the individual ID_50_ values for the Delta variant and wild-type variant for each subject. Asterisks indicate significance levels: *** *p* < 0.001. ID_50_: reciprocal value of the serum dilution that showed a 50% protection of virus cytopathic effect.

**Figure 3 vaccines-10-00580-f003:**
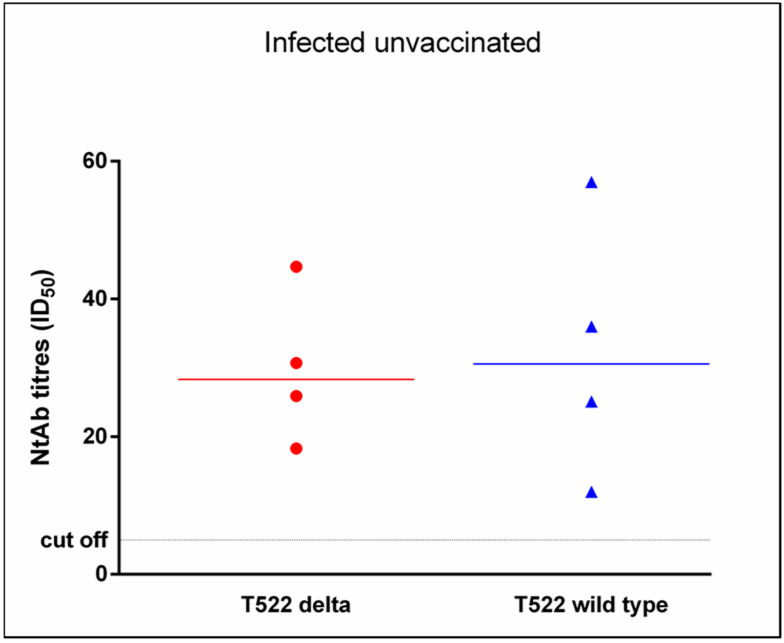
Neutralizing antibody titers for the wild-type lineage and the Delta variant in four naturally infected unvaccinated healthcare workers tested after a median of 522 days from infection. Data are reported as individual ID_50_ values and as median values. ID_50_: reciprocal value of the serum dilution that showed a 50% protection of virus cytopathic effect.

## Data Availability

The raw data can be made available by the corresponding author upon reasonable request.

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
