# Peer review of "Comparable Post-Vaccination Decay of Neutralizing Antibody Response to Wild-Type and Delta SARS-CoV-2 Variant in Healthcare Workers Recovered from Mild or Asymptomatic Infection"

_vaccines, 2022, doi:10.3390/vaccines10040580_

Round 1

Reviewer 1 Report

The communication article “Comparable post-vaccination decay of neutralizing antibody response to wild type and delta SARS-CoV-2 variant in health care workers recovered from mild or asymptomatic infection “submitted by Vicenti et al to the Vaccines Journal (vaccines-1650331) is quite interesting and appreciable.

However, a few things need to be clarified in the communication article:

  • After BNT162b2 mRNA vaccination, how many days sufficient level of neutralizing antibodies will exist in vaccinated individuals? What is the threshold level of NtAb?
  • How did the authors fix the interval time as T20 and T200 to test the NtAb?
  • What are the observed mild or asymptomatic infections/symptoms in selected individuals?
  • The authors need to include the unique nature of the delta variant in the discussion.
  • Whether the authors noticed any allergic responses in vaccinated individuals.
  • The authors need to discuss, the reasons for the post-vaccination decay of NtAb response. It might increase the readability and quality of the article.

Author Response

Response to Reviewer 1 Comments

The communication article “Comparable post-vaccination decay of neutralizing antibody response to wild type and delta SARS-CoV-2 variant in health care workers recovered from mild or asymptomatic infection “submitted by Vicenti et al to the Vaccines Journal (vaccines-1650331) is quite interesting and appreciable.

However, a few things need to be clarified in the communication article:
Point 1. After BNT162b2 mRNA vaccination, how many days sufficient level of neutralizing antibodies will exist in vaccinated individuals? What is the threshold level of NtAb?

Response 1: We included in the study previously infected subjects tested 20 and 200 days after the second BNT162b2 mRNA vaccine: a detectable level of NtAb at the longest interval, similar to our findings, is expected (Herzberg et al., Persistence of Immune Response in Health Care Workers After Two Doses BNT162b2 in a Longitudinal Observational Study. Front Immunol. 2022;13:839922; Benning et al. Neutralizing antibody activity against the B.1.617.2 (delta) variant 8 months after two-dose vaccination with BNT162b2 in health care workers. Clin Microbiol Infect. 2022, doi: 10.1016/j.cmi.2022.01.011). Technically, the sensitivity threshold level to consider NtAb as positive and neutralizing is ID50 titres ≥10, as reported in Materials and Methods section. However, to establish a threshold for in vivo protection is a complex issue because several factors must be considered including T cell immunity and containment measures as well as selection biases impacting exposure to virus in different population studies. A recent study defined the median threshold for in vivo protection at 154 Binding Antibody Units (BAU) in a spike RBD binding assay with the ancestral wild type virus (Goldblatt et al, Towards a population-based threshold of protection for COVID-19 vaccines.” Vaccine 2022; 40;306-315.), however establishing a correspondence with the different neutralization assays remains tricky.
We included the explanation in the text.

Point 2. How did the authors fix the interval time as T20 and T200 to test the NtAb?

Response 2: Thanks for the question. T20 was chosen as the peak time for antibody detection based on our previous study (Vicenti et al, Faster decay of neutralizing antibodies in never infected than previously infected healthcare workers three months after the second BNT162b2 mRNA COVID-19 vaccine dose. Int J Infect Dis. 2021;112:40-44), while T200 was the longest interval before the start of third vaccine dose administration, which was mandatory for HCWs.

Point 3. What are the observed mild or asymptomatic infections/symptoms in selected individuals?

Response 3: Most HCWs with mild disease had fever (8 out of 10). We added this information in the text.

Point 4: The authors need to include the unique nature of the delta variant in the discussion.

Response 4: We agree with the Reviewer. The text was modified as follows in the Discussion section: “Delta VoC was first documented in India in December 2020. This VoC presents a unique mutation T478K in the spike glycoprotein with high binding affinity with ACE2: along with L452R, it provides immune escape (Kumar et al, J Med Virol. 2022;94:1641-1649; Liu et al Identification of SARS-CoV-2 Spike mutations that attenuate monoclonal and serum antibody neutralization. Cell Host Microbe. 2021, 29, 477–488.e4; Perez-Gomez et al, The Development of SARS-CoV-2 Variants: The Gene Makes the Disease. J. Dev. Biol. 2021, 9, 58.). The clinical relevance is related to the faster replication, the higher contagiousness before symptoms onset and the occurrence of superspreading events, which make contact tracing more difficult (Lewis et al, Superspreading drives the COVID pandemic—And could help to tame it. Nature 2021, 590, 544–546; Li et al,
Viral infection and transmission in a large, well-traced outbreak caused by the SARS-CoV-2 Delta variant. Nat Commun. 2022;13:460): furthermore hospitalization rates are higher compared to pre-Delta variants (Twohig et al, Hospital admission and emergency care attendance risk for SARS-CoV-2 delta (B.1.617.2) compared with alpha (B.1.1.7) variants of concern: a cohort study. Lancet Infect Dis. 2022;22:35-42)”.

Point 5: Whether the authors noticed any allergic responses in vaccinated individuals.

Response 5: No allergic response occurred.
We added this sentence in Material and Methods section: “No allergic response to vaccination occurred”.

Point 6: The authors need to discuss, the reasons for the post-vaccination decay of NtAb response. It might increase the readability and quality of the article.

Response 6: We agree with the Reviewer. The text was modified as follows: “The level of neutralizing antibodies reflects humoral immune response against SARS-CoV-2 infection and BNT162b2 vaccine has a very large effect on it (Rogliani et al, SARS-CoV-2 Neutralizing Antibodies: A Network Meta-Analysis across Vaccines. Vaccines 2021, 9, 227): however, both in case of natural infection elicited by wild type strain and in case of vaccination, VoC may evade immune responses. For Delta, which bears key immune escape mutations such as L452R, P681R, and T478K, a substantial fold reduction of neutralizing antibodies elicited by vaccination or infection compared with reference lineages was previously reported (Chen et al, Neutralizing Antibodies Against Severe Acute Respiratory Syndrome Coronavirus 2 (SARS-CoV-2) Variants Induced by Natural Infection or Vaccination: A Systematic Review and Pooled Analysis. Clin Infect Dis. 2022;74:734-742).

Reviewer 2 Report

Manuscript Title: Comparable post-vaccination decay of neutralizing antibody response to wild type and delta SARS-CoV-2 variant in health care workers recovered from mild or asymptomatic infection

Reviewer Recommendation: Minor Revision

  1. Is the duration and limited number of cases in the current study sufficient for the analysis of the NtAb on HCWs with COVID-19 infection?
  2. Provide the details of the number of volunteers that have been utilized for this study? 
  3. Do the HCWs who were selected for this study have any comorbidity? Did authors have checked all these details for this study?
  4. The detailed abbreviations of some short forms are not provided in the manuscript and it makes reading the manuscript difficult.
  5. In figure 1, what does the colored symbol mean? If authors provide the information in more detail it could be more informative. 

Author Response

Response to Reviewer 2 Comments

Reviewer Recommendation: Minor Revision

Point 1: Is the duration and limited number of cases in the current study sufficient for the analysis of the NtAb on HCWs with COVID-19 infection?

Response 1: We are aware of the low number of subjects enrolled but, on the other hand, they had the peculiar characteristics to be regularly screened for SARS-CoV-2 infection since the pandemic start, so that both the timing of infection is known and we are sure that no reinfection occurred. In our opinion, the duration of the study is sufficient because it was the longest possible interval before the mandatory third vaccine dose.

Point 2: Provide the details of the number of volunteers that have been utilized for this study?

Response 2: We agree with the Reviewer about the need of greater clarity: 13 HCWs were asymptomatic (M/F 3/10) and 10 had mild disease (M/F 5/5). We included this information in the text.

Point 3: Do the HCWs who were selected for this study have any comorbidity? Did authors have checked all these details for this study?

Response 3: Thanks for the question. All HCWs were immunocompetent: two of them had uncomplicated arterial hypertension. We added this information in material and methods section.

Point 4: The detailed abbreviations of some short forms are not provided in the manuscript and it makes reading the manuscript difficult.

Response 4: We apologize, we modified the manuscript where necessary.

Point 5: In figure 1, what does the colored symbol mean? If authors provide the information in more detail it could be more informative.

Response 5: The legend was modified as follows: “The same colored symbol indicate the individual ID50 values in the same subject at different time points”.

Reviewer 3 Report

Major corrections:

Line 73-75: I am not sure it is clear. Did the authors perform a non-parametric paired test (of the same sample) for the two strains (B.1 and Delta) for 20 days? The same for 200 days? From figure 1, I get the impression that the median value for T20-Delta is lower than T20-wildtype.

Line 103-105: This statement needs to be rephrased and followed by much more discussion. Indeed, reduced immune response after some time plays a role, but the genomic characteristics of each VoC and its mutations are also important for immune evasion. Please check and thoroughly discuss the Cameroni et al paper (doi: 10.1038/s41586-021-04386-2).

The authors need to further discuss this very important issue of how mutations affect immune-evasion, in general.

So far, the spike-focused vaccines have higher efficacies compared to the inactivated virus vaccines (https://doi.org/10.1101/2021.09.17.21263549). However, they are also vulnerable to mutations, because the Spike is a well documented region of known genetic instability (https://doi.org/10.3390/v14010078 and doi: 10.1093/molbev/msab292). Spike mutations have previously led to the emergence of VoCs that are already capable of partly evading the immunity conferred by current vaccines (doi: 10.1016/j.cell.2021.03.013, doi: 10.1016/j.cell.2021.02.037).

Therefore, the author’s findings contradict many other previously published results. Thus, they should be very cautious in how they discuss their very controversial findings. They should also mention at the end of the manuscript that a higher number of samples, or independent studies would be needed to verify their controversial findings.

Minor corrections:

Line 15: antigen stimulations

Line 18-20: please rephrase

Iine 31: Until December

Line 35: until December 2021.

Line 53-54: When did the HCWs receive their first dose of the vaccine? Around the same time?

Line 92: Please correct this, concerning symbols and individuals.

Line 95-98: This paragraph needs rephrasing and a corresponding figure.

Author Response

Response to Reviewer 3 Comments

Major corrections
Point 1: Line 73-75: I am not sure it is clear. Did the authors perform a non-parametric paired test (of the same sample) for the two strains (B.1 and Delta) for 20 days? The same for 200 days? From figure 1, I get the impression that the median value for T20-Delta is lower than T20-wildtype.

Response 1: As reported in the material and methods section “The non-parametric Wilcoxon Signed Rank Sum test and Mann-Whitney test were used to analyze changes in paired and unpaired data, respectively”. Median value for T20-Delta is lower than T20-wildtype but the difference was not statistically significant: we agree with the Reviewer about the need of greater clarity and the text was modified as follows: “At 20 days, median (IQR) NtAb titers against delta virus was lower than titers against wild type (1432.0 [1188.9-2277.6] vs 1837.6 [1449.2-2426.7] but the difference was not statistically significant (p=0.3070)”.

Point 2: Line 103-105: This statement needs to be rephrased and followed by much more discussion. Indeed, reduced immune response after some time plays a role, but the genomic characteristics of each VoC and its mutations are also important for immune evasion. Please check and thoroughly discuss the Cameroni et al paper (doi: 10.1038/s41586-021-04386-2).

Response 2: we agree with the Reviewer, the original sentence was modified as follows: “The lack of long-term SARS-CoV-2 immunity may explain the surge of SARS-CoV-2 delta infections in vaccinated people but other hypotheses, as a role for delta specific escape from vaccine induced immunity need to be taken into account: mutations in the main antigenic sites of the Delta spike have the ability to evade recognition by monoclonal antibodies (McCallum et al,Molecular basis of immune evasion by the Delta and Kappa SARS-CoV-2 variants. Science. 2021, 374, 1621-1626). Different genomic characteristics of VoC had a clinical relevance: many of the Omicron RBD mutations are found at positions which are important contact sites with ACE2 and Cameroni et al (Cameroni et al, Broadly neutralizing antibodies overcome SARS-CoV-2 Omicron antigenic shift. Nature. 2022, 602, 664-670.) reported that Omicron RBD binds to human ACE2 with increased affinity”

Point 3: The authors need to further discuss this very important issue of how mutations affect immune-evasion, in general.
So far, the spike-focused vaccines have higher efficacies compared to the inactivated virus vaccines (https://doi.org/10.1101/2021.09.17.21263549). However, they are also vulnerable to mutations, because the Spike is a well documented region of known genetic instability (https://doi.org/10.3390/v14010078 and doi: 10.1093/molbev/msab292). Spike mutations have previously led to the emergence of VoCs that are already capable of partly evading the immunity conferred by current vaccines (doi: 10.1016/j.cell.2021.03.013, doi: 10.1016/j.cell.2021.02.037).
Therefore, the author’s findings contradict many other previously published results. Thus, they should be very cautious in how they discuss their very controversial findings. They should also mention at the end of the manuscript that a higher number of samples, or independent studies would be needed to verify their controversial findings.

Response 3: We agree with the Reviewer, we added the following text to the Discussion: “BNT162b2 vaccine efficacy estimates after a two-dose regimen was up to 99% for death and 100% for severe COVID-19 [Higdon et al, A systematic review of COVID-19 vaccine efficacy and effectiveness against SARS-CoV-2 infection and disease. medRxiv 2021.09.17.21263549]: head-to-head comparisons reported a slightly lower protective effect for symptomatic disease or any infection caused by Delta compared to Alpha (Sheikh et al, SARS-CoV-2 Delta VOC in Scotland: demographics, risk of hospital admission, and vaccine effectiveness. Lancet. 2021;397:2461-2462).
The spike protein is responsible for binding to host receptors, a key function driving evolution towards highly human-adapted phenotypes via point mutations and recombination events at higher rates than any other part of the viral genome (Amoutzias et al, The Remarkable Evolutionary Plasticity of Coronaviruses by Mutation and Recombination: Insights for the COVID-19 Pandemic and the Future Evolutionary Paths of SARS-CoV-2. Viruses 2022, 14, 78; Nikolaidis et al, The Neighborhood of the Spike Gene Is a Hotspot for Modular Intertypic Homologous and Nonhomologous Recombination in Coronavirus Genomes. Mol Biol Evol. 2022, 39, msab292). Many VoC arose at the end of 2020, with mutations spanning the entire protein but mostly in S1 and RBD, which is the main target of neutralizing antibodies. However, also non-RBD mutations are involved in escape from vaccine-induced humoral immunity and continuous monitoring of the emergence of new variants is warranted together with detection of breakthrough infections (Garcia-Beltran et al, Multiple SARS-CoV-2 variants escape neutralization by vaccine-induced humoral immunity. Cell. 2021, 184, 2372-2383.e9; Zhou et al, Evidence of escape of SARS-CoV-2 variant B.1.351 from natural and vaccine-induced sera. Cell. 2021,184, 2348-2361.e6)
We included the limits of the study and the text was modified as follows: “Several previously published studies reported a lower vaccine-induced cross-neutralizing activity against the Delta variant compared to the wild type strain: our apparently contrasting results may have been influenced by the low numerosity of HCWs tested, thus further studies are needed to confirm our data.”

Minor corrections:
Point 4: Line 15: antigen stimulations

Response 4: the text was modified as requested

Point 5: Line 18-20: please rephrase.

Response 5: the text was modified as follows: “Median NtAb titers against B.1 and Delta are comparable and highly correlated at both 20 and 200 days after the second vaccine dose in the 23 HCWs enrolled (median age 46 years).”

Point 6: Iine 31: Until December

Response 6: the text was modified as requested

Point 7: Line 35: until December 2021.

Response 7: the text was modified as requested

Point 8: Line 53-54: When did the HCWs receive their first dose of the vaccine? Around the same time?

Response 8: The HCWs received their first dose after a median of 296 days after diagnosis of infection, as reported in the Results section. We added the statement “and the HCWs were all vaccinated in one-month period”.

Point 9: Line 92: Please correct this, concerning symbols and individuals.

Response 9: We agree with the Reviewer, the text was modified as requested: “The coloured symbol indicates the individual ID50 values for Delta variant and wild type variant for each subject”.

Point 10: Line 95-98: This paragraph needs rephrasing and a corresponding figure.

Response 10: We agree with the Reviewer: the text was modified as follows: “Interestingly, median NtAb titres to wild type and to Delta were comparable (30.5 [18.5-46.5] vs. 28.3 [22.1-37.7], p=1) also in four naturally
infected unvaccinated HCWs: this result suggests that a comparable response to these different variants is independent from vaccination” and Figure 3 was added.

Round 2

Reviewer 3 Report

The authors have addressed my corrections/concerns.

There are several typos across the manuscript. They need to re-check with a native english speaker.

Some of them bellow:

Delta or delta throughout the text.

Line 18: ..

Line 38: antibodies

Line 38: in vitro (italicize)

Line 56: underwent regular

Line 66: were used

Line 96:..

Line 134: in vivo (italicize)

Author Response

Response to Reviewer 3 Comments

Point 1: The authors have addressed my corrections/concerns.

There are several typos across the manuscript. They need to re-check with a native english speaker.
Some of
them bellow.

Response 1: We apologize for the typos and we agree with the Reviewer about the need to re-check with a
native english speaker. The manuscript was submitted to MDPI for English editing (English editing ID:
English-42489).

Point 2: Delta or delta throughout the text.

Response 2: the form “Delta” was applied throughout the text.

Point 3: Line 18: ..

Response 3: The text was modifie4d as requested.

Point 4: Line 38: antibodies
Response 4: The text was modifie4d as requested.

Point 5: Line 38: in vitro (italicize)

Response 5: The text was modifie4d as requested.

Point 6: Line 56: underwent regular

Response 6: The text was modifie4d as requested.

Point 7: Line 66: were used

Response 7: The text was modified as requested.

Point 8: Line 96:..

Response 8: The text was modified as requested.

Point 9: Line 134: in vivo (italicize)

Response 9: The text was modified as requested.